# Fault Feature Extraction Method for Rolling Bearings Based on Complete Ensemble Empirical Mode Decomposition with Adaptive Noise and Variational Mode Decomposition

**DOI:** 10.3390/s23239441

**Published:** 2023-11-27

**Authors:** Lijing Wang, Hongjiang Li, Tao Xi, Shichun Wei

**Affiliations:** 1School of Control and Mechanical Engineering, Tianjin Chengjian University, Tianjin 300384, China; lijing8668@tcu.edu.cn (L.W.); 15222521538@163.com (H.L.); 18822593936@163.com (S.W.); 2School of Mechanical Engineering, Tiangong University, Tianjin 300387, China

**Keywords:** rolling bearing, fault feature extraction, CEEMDAN, VMD, SSA

## Abstract

Due to the difficulty in dealing with non-stationary and nonlinear vibration signals using the single decomposition method, it is difficult to extract weak fault features from complex noise; therefore, this paper proposes a fault feature extraction method for rolling bearings based on complete ensemble empirical mode decomposition with adaptive noise (CEEMDAN) and variational mode decomposition (VMD) methods. CEEMDAN was used to decompose the signal, and the signal was then screened and reconstructed according to the component envelope kurtosis. Based on the kurtosis of the maximum envelope spectrum as the fitness function, the sparrow search algorithm (SSA) was used to perform adaptive parameter optimization for VMD, which decomposed the reconstructed signal into several IMF components. According to the kurtosis value of the envelope spectrum, the optimal component was selected for an envelope demodulation analysis to realize fault feature extraction for rolling bearings. Finally, by using open data sets and experimental data, the accuracy of envelope kurtosis and envelope spectrum kurtosis as a component selection index was verified, and the superiority of the proposed feature extraction method for rolling bearings was confirmed by comparing it with other methods.

## 1. Introduction

As one of the main parts of rotating machinery [1,2,3,4], rolling bearings are widely used in high-end manufacturing, including aviation [5,6], automobiles [7,8], and ships [9,10]. However, influenced by load, installation, and lubrication factors, a variety of failures appear [11,12], which are easily overwhelmed by complex noise in a complex and changeable working environment. Thus, effective fault features cannot be directly extracted [13,14,15]. Therefore, separating and identifying fault features in complex noise can effectively prevent great economic losses and assure the safe operation of the devices.

As an effective method, vibration signals are often analyzed to obtain fault information during fault feature extraction. However, the single decomposition method is used to effectively extract weak fault features from non-stationary and nonlinear vibration signals, whose noise components are complex in actual working conditions [16]. Recently, signal decomposition methods, such as the short-time Fourier transform (STFT) [17,18], discrete wavelet transform (DWT) [19,20], and ensemble empirical mode decomposition (EEMD) [21] methods, have been introduced. In addition, in the field of fault diagnosis, the multivariate statistical analysis method is also an important research tool that has been widely applied for fault diagnosis purposes [22,23]. However, the decomposition effect of the STFT is affected by the window size and fixed time–frequency resolution. The DWT can be disturbed by the cross-term, which can have an influence on the original signal. Wu et al. [21] proposed EEMD to add white noise to the analyzed signal and automatically separate different time scales’ signals into the corresponding reference scales. The white noise and the number of iterations depend on human experience in EEMD. Torres et al. [24] proposed the adaptive noise-complete set empirical mode decomposition (CEEMDAN) model by introducing positive and negative adaptive noise mechanisms; the decomposed IMF components were averaged, which reduced the error of the reconstructed signal created by the selected components.

Dragomiretskiy et al. [25] proposed the variational mode decomposition (VMD) method to construct a variation optimization model to extract different vibration modes from the signal. Compared with the traditional mode decomposition method, VMD solves the problem of the end effect and mode aliasing as a non-recursive decomposition mode. However, the decomposition effect of VMD on the signal is influenced by the preset parameters K and α [26,27]. Intelligent optimization algorithms have been introduced to adaptively search for the values of K and α [28,29]. For example, Zhou et al. [30] and Zhang et al. [31] used the ant colony algorithm (ACO) and grasshopper optimization algorithm (GOA), respectively, to optimize VMD parameters. Instead of using the abovementioned methods, which obtained optimal solutions too easily and converged slowly, Xue et al. [32] proposed the sparrow search algorithm (SSA) based on the behavior of the sparrow population when foraging and avoiding natural enemies. As a new optimization algorithm, the SSA has the advantages of a strong global optimization ability, good robustness, rapid convergence speed, and not easily falling into local optima.

The existence of noise affects the selection of the IMF, which makes it impossible to extract an ideal fault characteristic signal. Both Tan et al. [33] and Wei et al. [34] adopted the minimum envelope entropy (MEE) method to select the evaluation indicators of components. They both achieved good results. However, bearings often work in a complex and changeable environment, and the collected signal inevitably contains a lot of noise. It is difficult to accurately select fault components by using a single index; so, it is impossible to extract ideal fault features from reconstructed signals. Therefore, envelope kurtosis combined with envelope spectrum kurtosis is proposed in this paper as an index to select IMF components.

Based on the abovementioned analysis, the method proposed in this article uses CEEMDAN for noise reduction purposes for fault signals and selects the optimal component by using envelope kurtosis. Based on the kurtosis of the maximum envelope spectrum as the fitness function, the SSA is used to perform parameter optimization for VMD, which decomposes the reconstructed signal into several IMF components. According to the kurtosis value of the envelope spectrum, the optimal component is selected for the envelope demodulation analysis so as to realize fault feature extraction for rolling bearings despite complex noise interference.

## 2. Theoretical Introduction

### 2.1. CEEMDAN

As an extension of EMD [35], CEEMDAN [24] solves the mode aliasing problem in EMD by adding Gaussian white noise to the data to achieve iterative decomposition and obtain multiple sets of inherent modal functions, and the residual component ensures the integrity of the reconstructed signal by averaging the IMF components obtained from each order decomposition to improve the reconstruction rate.

With CEEMDAN, the process of decomposing a vibration signal is as follows.

Step 1: Gaussian white noise ±*ε*_1_(*t*) generated with equal probability is added to the original vibration signal *x*(*t*) to obtain signal *x*(*t*) *± ε*_1_(*t*), which EMD decomposes to obtain M components, *IMF*_1m_. Then, the first-order eigenmode component *IMF*_1_(*t*) is obtained by averaging *IMF*_1m_:(1)$$IMF_{1}(t)=\frac{1}{M}\sum_{m=1}^{M}IMF_{1m}(t)$$

Step 2: The first eigenmode component in *x*(*t*) is removed to obtain the first residual component *r*_1_(*t*):(2)$$r_{1}(t)=x(t)-IMF_{1}(t)$$

Step 3: Equal positive and negative Gaussian white noise ± *ε*_2_(*t*) is continuously added to *r*_1_(*t*) to obtain signal *r*_1_(*t*) ± *ε*_2_(*t*), which is decomposed into multiple components, *IMF*_2*m*_. Then, the second-order eigenmode component *IMF*_2_(*t*) is obtained by averaging *IMF*_2*m*_:(3)$$IMF_{2}(t)=\frac{1}{M}\sum_{m=1}^{M}IMF_{2m}(t)$$

Step 4: The second eigenmode component in *r*_1_(*t*) is removed to obtain the second residual component *r*_2_(*t*):(4)$$r_{2}(t)=r_{1}(t)-IMF_{2}(t)$$

Step 5: By analogy, the eigenmode component of order N + 1 is obtained until running to the N + 1 stage:(5)$$IMF_{n+1}(t)=\frac{1}{M}\sum_{m=1}^{M}IMF_{(n+1)m}(t)$$

Step 6: The above steps are repeated until the residual component is a monotone function. Then, the original vibration signal x(t) is the sum of N eigenmode components and the residual:(6)$$x(t)=\sum_{n=1}^{N}IMF_{n}(t)+R(t)$$

### 2.2. VMD

To avoid mode aliasing, VMD [25] forgoes the recursive solution idea used in the traditional signal decomposition algorithm, adopts an entirely non-recursive method to construct a constrained variational model, and decomposes the input signal into K IMF components. Then, the constrained variational model based on VMD uses the Hilbert transform to construct an analysis signal to obtain a unilateral spectrum, adopts a Wiener filter to denoise the signal, and moves the spectrum to the baseband through frequency mixing.

It can be expressed as follows:(7)$$\{\begin{array}{c} \underset{\left\{u_{k}\right\},\left\{\omega_{k}\right\}}{\min}\left\{\sum_{k}{\|\partial_{t}\left[\left(\delta (t)+\frac{j}{\pi t}\right)*u_{k}(t)\right]e^{-j\omega_{k}t}\|}_{2}^{2}\right\}\\ s.t.\sum_{k=1}^{K}u_{k}(t)=f(t) \end{array}$$
where *u_k_* is the *k*-th IMF component, *ω_k_* is the central frequency of the corresponding IMF component, and *δ*(*t*) is the Dirichlet function.

The quadratic penalty factor α and Lagrange multiplier λ are introduced into Equation (7), transforming it into an unconstrained variational problem.
(8)$$L(\{u_{k}\},\{\omega_{k}\},\lambda )=\alpha {\sum_{k=1}^{K}\left\|(\partial_{t}[\delta (t)+\frac{j}{\pi t})\times u_{k}(t)]\times e^{-j\omega_{k}t}\right\|}_{2}^{2}+{\left\|f(t)-\sum_{k=1}^{K}u_{k}\right\|}_{2}^{2}+\langle \lambda (t),f(t)-\sum_{k=1}^{K}u_{k}\rangle$$

The alternating direction multiplication operator iteratively updates *u_k_*, *ω_k_*, and *λ*.
(9)$${\hat{u}}_{k}^{n+1}(\omega )=\frac{\hat{f}(\omega )-\sum_{i=1}^{k-1}{\hat{u}}_{i}^{n}(\omega )+\frac{\hat{\lambda }(\omega )}{2}}{1+2\alpha {\left(\omega -\omega_{k}^{n}\right)}^{2}}$$
(10)$$\omega_{k}^{n+1}=\frac{{\int_{0}^{\infty }\omega \left|{\hat{u}}_{k}^{n+1}(\omega )\right|}^{2}d\omega }{{\int_{0}^{\infty }\left|{\hat{u}}_{k}^{n+1}(\omega )\right|}^{2}d\omega }$$
(11)$${\hat{\lambda }}^{n+1}(\omega )={\hat{\lambda }}^{n}(\omega )+\tau (\hat{u}(\omega )-\sum_{k=1}^{K}{\hat{y}}_{k}^{n+1}(\omega ))$$

The signal decomposition effect of VMD is affected by the parameters *K* and *α*. When the value of *K* is too large, the signal may be over-decomposed, resulting in signals that should be in the same frequency band being decomposed into multiple bands. When the value of *K* is too small, the signal may be under-decomposed, resulting in components that should be in different frequency bands being decomposed into the same frequency band. When *α* is too large, the bandwidth between modes becomes smaller, which quickly causes information to be lost. When the *α* value is too small, the bandwidth between the modes becomes more extensive, which quickly causes more noise to be retained [36].

### 2.3. VMD Parameter Optimization Based on SSA

As a swarm intelligent optimization algorithm based on the behavior of sparrows foraging and escaping predators, the SSA [30] is widely used due to its few adjustment parameters, simple implementation, and fast calculation speed. Sparrows in the population are divided into discoverers and followers. Discoverers have better fitness values than followers, allowing them to obtain food faster when foraging. Then, some individuals in the sparrow population are selected as watchmen responsible for guarding. When danger is detected, the sparrow population stops foraging and moves to a safe place [37].

The process of the adaptive selection of VMD parameters by SSA is as follows:(1)Initialization of SSA parameters: The sparrow number is set to *n* = 20, with maximum iterations *T* = 10 and the safety value *ST* = 0.5. The number of discoverers accounts for 70% of the sparrow population, the number of participants accounts for 30%, and 20% of the sparrows are selected from discoverers and followers as the watchmen.(2)The positions of sparrows in the population are set as X = $\left[\begin{array}{cccc} x_{1,1} & x_{1,2} & \ldots & x_{1,d}\\ x_{2,1} & x_{2,2} & \ldots & x_{2,d}\\ \ldots & \ldots & \ldots & \ldots \\ x_{n,1} & x_{n,2} & \ldots & x_{n,d} \end{array}\right]$, where d is the variable dimension of the optimization problem. Then, the fitness values of all sparrows can be expressed as F_X_ = $\left[\begin{array}{c} f([\begin{array}{cccc} x_{1,1} & x_{1,2} & \ldots & x_{1,d} \end{array}])\\ f([\begin{array}{cccc} x_{2,1} & x_{2,2} & \ldots & x_{2,d} \end{array}])\\ \ldots \\ f([\begin{array}{cccc} x_{n,1} & x_{n,2} & \ldots & x_{n,d} \end{array}]) \end{array}\right]$. Sparrows are divided into discoverers and followers by ranking the fitness value of each sparrow. The current position of the sparrow with the optimal fitness value is the optimal position X_best_.(3)Compared to followers, discoverers need to obtain food faster during foraging but also undertake the task of helping sparrow groups find food and provide food directions to followers. Thus, the discoverers have a more comprehensive range of searches than followers. The position of the discoverers is iteratively updated according to the following equation:
(12)$$x_{i,j}^{t+1}=\{\begin{array}{ll} x_{i,j}^{t}\cdot \exp(\frac{-i}{\mu \cdot T}) & R_{2}\;<\;ST\\ x_{i,j}^{t}+Q\cdot L & R_{2}\geq ST \end{array}$$where $x_{i,j}^{t}$ is the *j*-th dimensional position of the *i*-th sparrow in the *t*-th iteration, *μ* is a random number between 0 and 1, *T* is the maximum number of iterations, *Q* is a random number that obeys a normal distribution, *L* is a 1 × *d* matrix with all 1 elements, *R*_2_ is a warning value between 0 and 1, and *ST* is the safety value between 0.5 and 1.

During the foraging process, some followers always observe discoverers. Once they realize that the discoverers have found better food, they leave their current position to compete for food. Then, the position of the participant is iteratively updated according to the following equation:(13)$$x_{i,j}^{t+1}=\{\begin{array}{ll} x_{i,j}^{t}\cdot \exp(\frac{x_{w}-x_{i,j}^{t}}{i^{2}}) & i>\frac{n}{2}\\ x_{p}^{t+1}+\left|x_{i,j}^{t}-x_{p}^{t+1}\right|\cdot A^{+}\cdot L & i\leq \frac{n}{2} \end{array}$$
where *x_w_* represents the worst position in the population, *x_p_* represents the optimal position occupied by the discoverer in the population, and A is the matrix of 1 × *j* consisting of 1 and −1.

When the sparrow population is searching for food, the watchmen mainly ensure the safety of the population. Then, the position of the watchman is iteratively updated according to the following equation:(14)$$x_{i,j}^{t+1}=\{\begin{array}{ll} x_{\mathrm{best}}+\beta \cdot \left|x_{i,j}^{t}+x_{\mathrm{best}}^{t}\right| & f_{i}>f_{g}\\ x_{i,j}^{t}+K\cdot (\frac{\left|x_{i,j}^{t}+x_{w}^{t}\right|}{(f_{i}-f_{w})+z}) & f_{i}=f_{g} \end{array}$$
where *x_best_* is the optimal location for the current population, *β* is the step control parameter, *K* is *a* random number between −1 and 1, *f_i_* is the fitness value of the current individual, *f_g_* and *f_w_* are the best and worst global fitness values, respectively, and *z* is a constant.

(4)During position iteration, the current optimal position X_best_ is obtained by updating the fitness ranking of sparrow individuals.(5)Steps (3) and (4) are repeated until meeting the maximum number of iterations, *T* = 10. Then, the optimal location X_best_ is determined and outputs the parameters *K* and *α*. Otherwise, the process returns to step (3).

When the kurtosis of the fitness function does not change in successive iterations, we can conclude that the current optimal solution is the final optimal solution. However, to avoid temporarily falling into local optima, we do not choose this as the criterion for stopping iteration but wait until the maximum number of iterations is met. The process of the adaptive selection of VMD parameters by the SSA is shown in Figure 1.

## 3. Proposed Methods

### 3.1. Component Selection Index

Two features should be considered in the fault feature extraction of rolling bearings: the fault signal has impact characteristics in the time domain and cyclic stationarity in the frequency domain. Then, the collected signal of rolling bearings is usually accompanied by impact noise and cyclic stationary noise in complex and changeable environments. This will affect the accuracy of the fault feature extraction of the rolling bearing. The impact of the fault signal can be expressed by the amplitude in the signal envelope, which is negligible most of the time. When the signal contains fault information, the amplitude will change considerably. Kurtosis is the most suitable index to measure the amplitude change in the signal. More significant kurtosis means more vibration with large values in the signal. Therefore, the signal envelope is combined with the kurtosis, and the greater the kurtosis of the envelope, the more fault information in the signal. The envelope kurtosis of a signal is determined by passing the signal through the Hilbert transform and then calculating the kurtosis of the transformed signal. The envelope kurtosis of the signal *K_H_* is as follows.
(15)$$K_{H}=\frac{\sum_{t=1}^{N}{\left(H(t)-Y_{H}\right)}^{4}}{\sigma_{H}^{4}}$$
where *H*(*t*) is the signal obtained in the envelope of the fault signal obtained by the Hilbert transform, *Y_H_* is the average of *H*(*t*), and *σ_H_* is the standard deviation of *H*(*t*).

The cyclic stationarity of the fault signal can be expressed by the amplitude of the envelope spectrum, which can identify the components of the frequency in the signal and their amplitude and phase information. When the signal contains a fault, the envelope spectrum will have a high amplitude of the components representing the fault frequency. The kurtosis can still represent the amplitude variation in the envelope spectrum. So, the larger the kurtosis of the envelope spectrum, the more faults the signal has. The kurtosis of the envelope spectrum of the signal is caused by the signal first passing through the Hilbert transform and then the Fourier transform to obtain the envelope spectrum. Then, the kurtosis of the envelope spectrum is calculated. The envelope spectrum kurtosis of the signal *K_F_* is as follows.
(16)$$K_{F}=\frac{\sum_{t=1}^{N}{\left(F(H(t))-Y_{F}\right)}^{4}}{\sigma_{F}{}^{4}}$$
where *F*(*H*(*t*)) is the signal after the *H*(*t*) Fourier transform, *Y_F_* is the average of *F*(*H*(*t*)), and *σ_F_* is the standard deviation of *F*(*H*(*t*)).

### 3.2. Feature Extraction Model Based on CEEMDAN and SSA-VMD

The fault signal was decomposed and denoised by CEEMDAN, combined with the SSA, to achieve an adaptive selection method for the optimal parameter combination for VMD to extract fault features. The process is shown in Figure 2. The steps are as follows.

Step 1: CEEMDAN was used to decompose the original vibration signal. By calculating the envelope kurtosis values of the original signal and each IMF component, the IMF component whose envelope kurtosis was greater than the original signal was reconstructed to generate a new vibration signal.

Step 2: The optimization range of the mode decomposition number *K* and the quadratic penalty factor *α* was set in VMD.

Step 3: The maximum envelope spectrum kurtosis value of IMF decomposed by VMD was used as the fitness function in SSA optimization, and the optimal parameter combination of *K* and *α* under different working conditions was obtained.

Step 4: The IMF component with the largest kurtosis of the envelope spectrum was selected for envelope demodulation analysis using the optimized VMD to decompose and reconstruct the signal.

## 4. Experiment

The experiment described in this section used public data sets from Case Western Reserve University (CWRU) and data obtained from a rolling bearing simulation test bed to verify the accuracy of component selection indicators and the validity of the fault feature extraction model based on CEEMDAN and SSA-VMD.

### 4.1. Open Data Sets from Case Western Reserve University

#### 4.1.1. Experimental Apparatus and Open Data Sets

The bearing fault simulation test rig published by CWRU [38] is shown in Figure 3. The #6205-2RS JEM SKF deep groove ball bearing was chosen as the test bearing at the drive end of a motor with a sampling frequency of 12 kHz and a rotating speed of 1730 r/min. This study chose single-point damage caused by electrical discharge machining as the fault type. The vibration signals of three kinds of bearing faults (inner-ring fault, outer-ring fault, and rolling-element fault) were collected. Then, the time-domain diagram of each fault type was drawn (Figure 4).

The vibration data in the data set are almost unaffected by noise interference to get closer to the vibration signal in the complex environment. According to a study of the literature [39], impact noise caused by electromagnetic interference, cyclic stationary noise caused by axis rotation, and Gaussian white noise were added to the vibration data, as shown in Figure 5. The noisy data y(t) can be expressed as follows:(17)$$y(t)=\underset{impact\;noise}{\underset{︸}{\sum_{j}R_{j}\cdot S(t-r_{j})}}+\underset{cyclic\;stationary\;noise}{\underset{︸}{\sum_{l}A_{l}\cos(2\pi f_{r}t+\theta_{l})+\sum_{k}B_{k}\cos(2\pi Rf_{r}t+\theta_{k})}}+\underset{Gaussian\;white\;noise}{\underset{︸}{n(t)}}$$
where *f_r_* is the frequency conversion, and *S*(*t*) is the pulse response function and calculated by using the following equation:(18)$$S(t)=\{\begin{array}{cc} \exp(-\lambda t)\sin(2\pi f_{o}t) & t>0\\ 0 & t\leq 0 \end{array}$$

The parameters of interference noise in Equations (17) and (18) are shown in Table 1.

Three kinds of interference noise (Figure 5) with different multiples were added to Figure 4, as shown in Figure 6. It can be seen in Figure 6 that the original signals were overwhelmed by the three kinds of interfering noise and could not display apparent fault features. A reasonable fault impact is not displayed in any of the envelope spectra (Figure 7). According to the analysis, the above traditional analysis method was only suitable for cases with less noise interference. It could not effectively extract the fault features of rolling bearings in the presence of complex noise interference.

#### 4.1.2. Feature Extraction Based on CEEMDAN and SSA-VMD

This paper takes the bearing outer-ring fault signal plus noise as an example. The signal X in Figure 6b was decomposed using CEEMDAN. In the CEEMDAN decomposition process, it is necessary to add positive and negative pairs of Gaussian white noise to reduce the occurrence of mode aliasing. The value of the added white noise affects the accuracy of the decomposition. After many experiments, Gaussian white noise with a signal ratio of 0.2 was selected. The result is 13 IMF components, as shown in Figure 8.

Equation (15) was used to calculate the envelope kurtosis of the signal with added noise and each IMF component, as shown in Table 2. Components of signal X greater than 3.29 were selected to reconstruct the signal, i.e., IMF1 = 3.29, IMF2 = 3.53, IMF3 = 3.81, IMF5 = 8.35, and IMF13 = 3.66. As shown in Figure 9, the cyclic stationary noise was filtered out, but the impact noise still existed. In the envelope spectrum, the fundamental frequency of the fault appeared, but frequency doubling still interfered with the noise. So, it was difficult to effectively extract the fault features of the bearing signal when only using the CEEMDAN method.

With the maximum envelope spectrum kurtosis as the fitness function, the adaptive optimization of the parameters K and α of VMD was carried out, as shown in Figure 10. As for the setting of the maximum number of iterations, according to the optimization ability of the SSA for VMD’s optimal parameter combination and experimental proof, the maximum number of iterations is set to 10, which can consider both the optimization accuracy and the algorithm’s running time. It can be seen in Figure 10 whether the current number of iterations has obtained the optimal solution. Finally, the optimal parameter combination K = 4 and α = 2883 obtained by optimization were used for the VMD of the reconstructed signal, and four IMF components were obtained, as shown in Figure 11.

Equation (16) was used to calculate the envelope spectrum kurtosis of four IMF components, as shown in Table 3. The IMF3 component with the largest kurtosis spectrum was selected for envelope demodulation analysis. It can be seen in Figure 12 that the cyclic stationary noise was filtered out, the base frequency *f_o_* and frequency doubling of the fault characteristic frequency were displayed, and the spectral line was prominent.

Bearings with an inner-ring fault and a rolling-element fault were extracted fault features based on CEEMDAN and SSA-VMD. As shown in Figure 13, the envelope spectrum showed that the complex noise was filtered out, the base frequency and frequency doubling of the fault characteristic frequency were displayed, and the fault impact was prominent. An analysis of Figure 12 and Figure 13 indicates that the feature extraction method proposed in this paper can efficiently extract the fault features of rolling bearings despite the interference of strong noise.

#### 4.1.3. Comparison with Other Methods

To verify the effectiveness of this method, it was compared with other algorithms (1) by comparing the difference in the optimal component envelope spectra and (2) by calculating the consistency correlation coefficient (CCC) between the selected optimal component and the original signal without noise. The CCC combines the characteristics of the correlation coefficient and mean square error, which can reflect not only the correlation of the signal but also the error value. The larger the *P_c_* value, the stronger the correlation and the better the algorithm performance. The formula for *P_c_* is as follows:(19)$$P_{c}=1-\frac{E[{\left(A-B\right)}^{2}]}{\sigma_{A}^{2}+\sigma_{B}^{2}+(\mu_{A}-\mu_{B})}$$
where *A* and *B* are variables, *σ* is the standard deviation, and *μ* is the mean.

Using the method proposed in this paper, the effective components were screened according to envelope kurtosis, and the parameters of VMD were optimized by the SSA using the maximum envelope spectrum kurtosis as the fitness function. The method was compared with the commonly used envelope entropy to verify the effectiveness of the proposed component selection index.

The size of the envelope entropy can reflect the degree of confusion of the vibration signal. The noise will be more substantial when the vibration signal becomes more chaotic. The signal in Figure 6b was decomposed based on CEEMDAN. Then, components were selected according to envelope entropy to reconstruct the signal. According to the analysis of the time-domain diagram of the reconstructed signal (Figure 14a), the impact noise still existed, and the cyclic stationary noise was not completely removed. Analyzing the envelope spectrum of a reconstructed signal (Figure 14b) revealed a high-amplitude impact and no obvious fault characteristic information.

The SSA takes the minimum envelope entropy as the fitness function and obtains the optimal parameter combination of *K* and *α* for the VMD of the reconstructed signal. According to the analysis in Figure 15, no evident fault influence can be found in the envelope spectrum due to the improper selection of evaluation indicators.

By using the above method, feature extraction was carried out for bearings with inner-ring and rolling-element faults to draw the envelope spectra shown in Figure 16. Compared with Figure 13, the impact noise in Figure 16 is not completely filtered, resulting in a large amount of noise in the low-frequency part. The explanation for Figure 15 and Figure 16 is that the selected components did not contain fault information, so there was no reasonable fault impact in the envelope spectrum because of an inappropriate component selection index. The comparison showed that the component selection index proposed was suitable for the accurate fault feature extraction of rolling bearings under complex environmental backgrounds.

The above method processes the three groups of signals of each fault type and calculates the CCC between the optimal component and the original signal. The results are shown in Table 4.

As shown in Table 4, the CCC between the processed and original signals ranges from 0.0021 to 0.0325. There is still a large amount of noise in the signal that needs to be filtered out, resulting in poor consistency between the two signals. Combined with the envelope spectrum, this method is difficult to apply to fault feature extraction with a complex noise background.

To verify the effectiveness of the proposed CEEMDAN combined with VMD, the fault feature extraction method was compared with EEMD combined with wavelet threshold decomposition (WTD).

Based on EEMD, the signal of each of the three types of bearing faults in Figure 6 was decomposed into multiple IMF components, and selected components were used to reconstruct the signal according to the envelope kurtosis. Then, the WTD was used to decompose the reconstructed signal to extract fault features, whose envelope spectrum was obtained, as shown in Figure 17. A comparison with Figure 13 reveals that it was affected by cyclic stationarity noise and Gaussian white noise for the inner-ring fault, by Gaussian white noise for the outer-ring fault, and by both impact noise and Gaussian white noise for the rolling-element fault in Figure 17, which can lead to inaccurate fault feature extraction.

The above method processes the three groups of signals of each fault type and calculates the CCC between the optimal component and the original signal. The results are shown in Table 5.

As shown in Table 5, the CCC between the processed and original signals is between 0.0181 and 0.0363. Although it is slightly higher than the envelope entropy as the evaluation index, the signal still contains much noise, resulting in poor consistency between the two groups of signals. In addition, there is no effective fault feature frequency in the envelope spectrum, which proves that the method of EEMD combined with WTD cannot effectively extract fault features under a complex noise background.

The method proposed in this paper was applied to process the three kinds of faults. The CCC between the optimal component and the original signal was calculated, and the results are shown in Table 6.

As shown in Table 6, the consistency correlation coefficient between the optimal component selected after applying the method proposed in this article and the original signal is between 0.4613 and 0.7201, and the consistency between the two sets of signals is much higher than that obtained using the other two methods. By combining the precise fault feature frequency in the envelope spectrum, it can be proven that the proposed method is effective in extracting fault features in complex noise backgrounds.

### 4.2. Data from Rolling Bearing Test Bed

#### 4.2.1. Experimental Apparatus and Data Acquisition

The proposed feature extraction method was tested by using data collected from a rolling bearing test bed, shown in Figure 18, to verify the universality. The #6205 deep groove ball bearing was chosen as the test bearing with a sampling frequency of 50 kHz. The vibration samples collected by the test bed contained three kinds of faults: an inner-ring fault, an outer-ring fault, and a rolling-element fault.

After the original vibration data of three types of rolling bearing faults were collected by the acceleration sensor in Figure 18, 25,000 consecutive data points randomly intercepted were added as the noise of Equation (17). Due to background noise interference, more obvious fault features cannot be found in the time-domain diagram (Figure 19), and a reasonable fault characteristic frequency cannot be seen in the envelope spectrum (Figure 20).

#### 4.2.2. Feature Extraction Based on CEEMDAN and SSA-VMD

The time-domain signal X’ of the outer-ring fault with added noise (Figure 19b) was collected and decomposed based on CEEMDAN to obtain 14 IMF components, as shown in Figure 21. Equation (15) was used to calculate the envelope kurtosis of the signal with added noise and each IMF component, as shown in Table 7. Components of signal *X*’ greater than 3.13 were selected to reconstruct the signal, i.e., IMF1 = 9.49, IMF2 = 4.31, IMF3 = 3.93, IMF4 = 3.39, IMF7 = 3.77, IMF9 = 3.54, and IMF11 = 5.31, to draw the time-domain diagram and envelope spectrum, as shown in Figure 22. The cyclic stationary noise was filtered out, but the impact noise still existed in the time-domain diagram. Meanwhile, the spectral line of the fault feature frequency was still not found in the envelope spectrum (Figure 20).

With the maximum envelope spectrum kurtosis as a fitness function, the optimal parameter combination *K* = 5 and *α* = 897 was obtained using the SSA to optimize VMD, by which the reconstructed signal was decomposed, as shown in Figure 23. Equation (16) was used to calculate the envelope spectrum kurtosis of five IMF components, respectively, as shown in Table 8. The IMF5 component with the largest kurtosis spectrum was selected for envelope demodulation analysis, as shown in Figure 24. According to the analysis, there is an obvious failure impact in Figure 23, and the fault feature frequencies *f_o_*, 2*f_o_*, and 3*f_o_* are shown in Figure 24.

Based on CEEMDAN and SSA-VMD, bearings with an inner-ring fault and a rolling-element fault were extracted fault features, respectively. As shown in Figure 25, the envelope spectrum shows that the fault feature frequency and double frequency can be found for the inner-ring fault, and the feature frequency can be extracted for the rolling-element fault. An analysis of Figure 24 and Figure 25 shows that the proposed feature extraction method was suitable for the complex environmental background and could accurately extract the fault features of rolling bearings.

#### 4.2.3. Compared with Other Feature Extraction Methods

To verify the effectiveness of the proposed method, the envelope entropy was used as the component selection index to extract the fault features of the signal in Figure 19. An analysis of the envelope spectrum in Figure 26 showed that the impact noise was not wholly filtered out; the noise still covered fault information for the inner and outer-ring faults, and there were other noise frequencies for the rolling-element fault. This was due to the improper selection of selection indicators, which made it difficult to find a reasonable fault shock in the envelope spectrum and impossible to extract accurate fault features.

EEMD combined with the WTD was used to extract the features of the rolling bearing signal in Figure 19. Analyzing the envelope spectrum in Figure 27 shows that the impact components were complex for the inner-ring fault, an unknown impact existed for the outer-ring fault, and many noise frequencies existed for the rolling-element fault. These would lead to inaccurate fault feature extraction. Therefore, the above comparison also shows that the proposed fault feature extraction method was more suitable for the accurate feature extraction of rolling bearing faults with complex environmental backgrounds.

The above three methods were used to process the three groups of signals of each fault type, and the consistency correlation coefficient between the optimal component and the original signal was calculated. The results are shown in Table 9.

As shown in Table 9, for the processed test bench signals, the CCC between the optimal component of the method proposed in this article and the original signal ranges from 0.1809 to 0.3821. This range is much higher than 0.0083–0.0927 obtained with the envelope entropy method and 0.0125–0.0329 with the combination of the EEMD and WTD methods. At the same time, the envelope spectrum can provide the fault characteristic frequency. It is proved that the proposed method is effective in denoising and fault feature extraction with complex noise backgrounds.

## 5. Conclusions

A fault feature extraction method based on CEEMDAN and SSA-VMD is proposed to solve the difficulty of fault feature extraction for rolling bearings. Collected fault signals are usually accompanied by impact and cyclic stationary noise. Envelope kurtosis and envelope spectrum kurtosis are proposed as component selection indexes. CEEMDAN was used to decompose the fault signal into multiple IMF components and a residual, which enhanced the robustness and reliability of the decomposition. According to the envelope kurtosis, the practical components were selected for signal reconstruction to reduce the influence of cyclic stationarity noise. The maximum envelope spectrum kurtosis was used as the fitness function of the SSA, and the optimal parameter combinations of K and α of VMD under different working conditions were obtained. The optimized VMD was used to decompose the reconstructed signal into several IMF components. Then, the optimal components were selected according to envelope spectrum kurtosis for envelope demodulation analysis to reduce the impact noise’s impact and realize fault feature extraction from rolling bearings.

The proposed methods were compared with envelope entropy as a component selection index and EEMD combined with the WTD method using open data sets and experimental data. The time-domain diagram and envelope spectrum verified that the proposed methods can accurately solve the difficult problem of fault feature extraction with a complex environmental background. In addition, the method proposed in this article can provide readers with some ideas for using secondary noise reduction and targeted denoising in fault feature extraction in the presence of complex noise interference. Meanwhile, we are attempting to apply this method to other types of faults and are conducting further research.

## Figures and Tables

**Figure 1 sensors-23-09441-f001:**
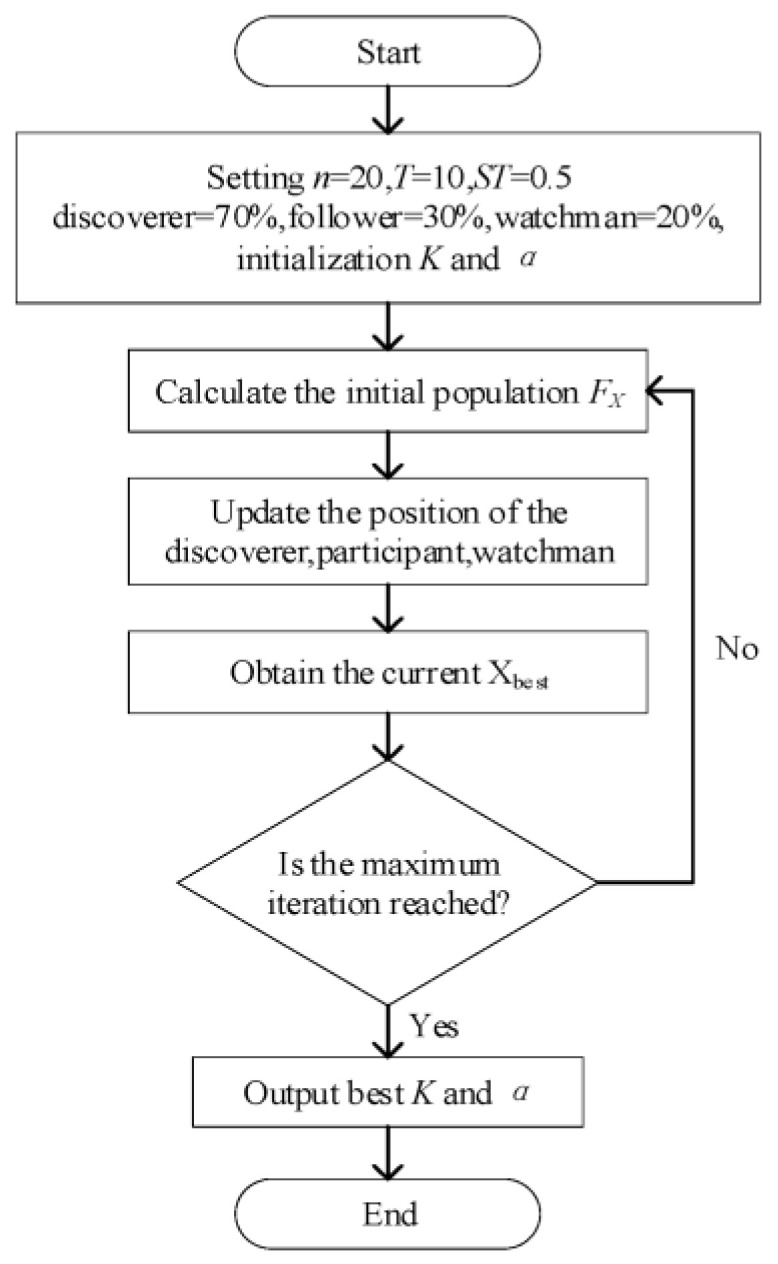
The flow diagram of SSA optimizing VMD parameters.

**Figure 2 sensors-23-09441-f002:**
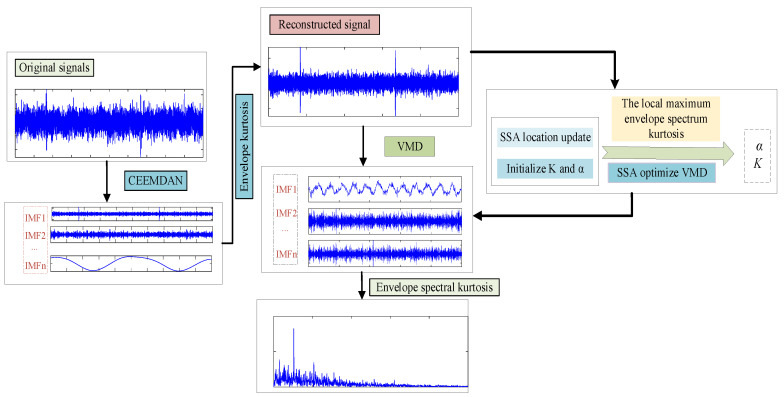
The flow diagram of fault feature extraction based on CEEMDAN and SSA-VMD.

**Figure 3 sensors-23-09441-f003:**
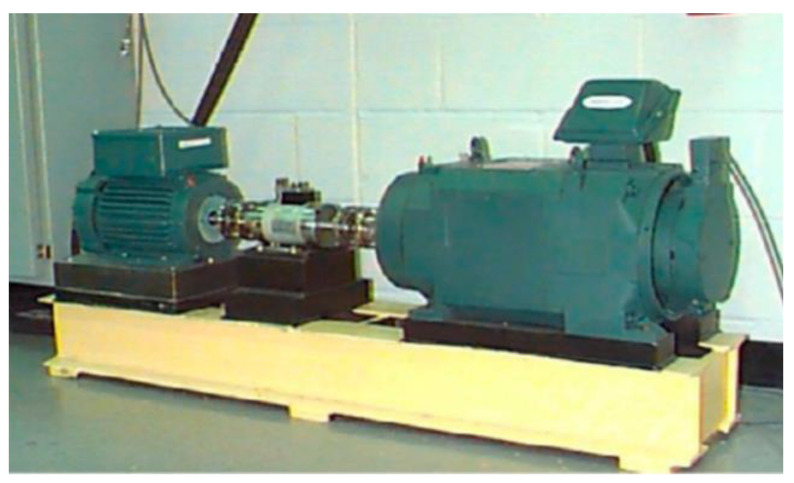
Rolling bearing failure test rig at CWRU.

**Figure 4 sensors-23-09441-f004:**
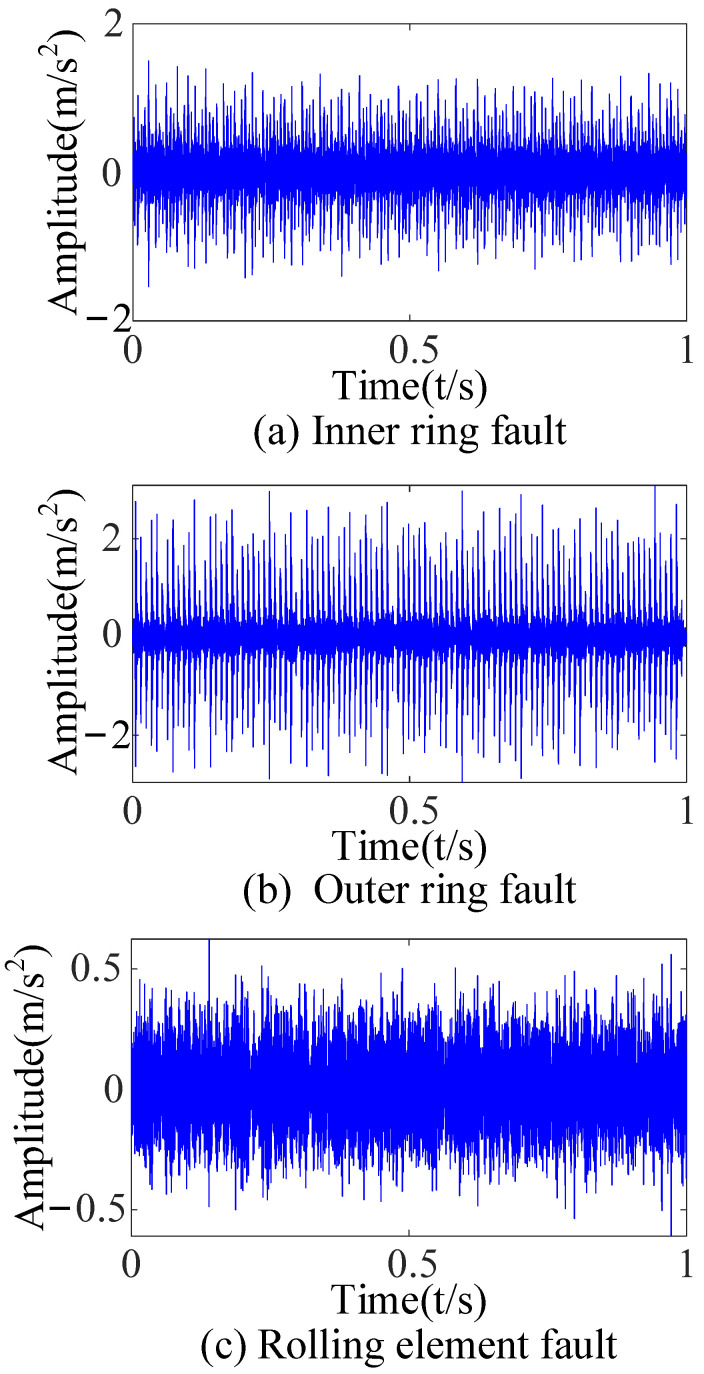
Time-domain diagrams of the original vibration signals.

**Figure 5 sensors-23-09441-f005:**
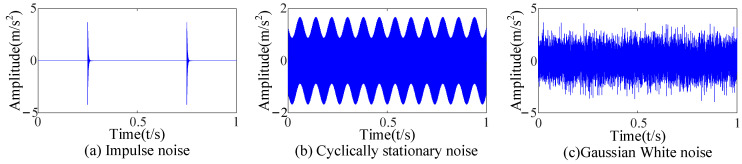
Time-domain diagrams of interference noise.

**Figure 6 sensors-23-09441-f006:**
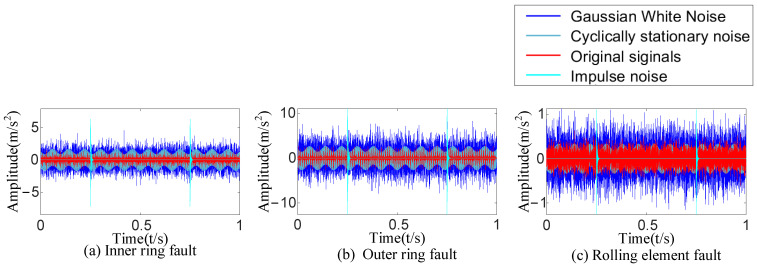
Noise data of three kinds of bearing faults.

**Figure 7 sensors-23-09441-f007:**
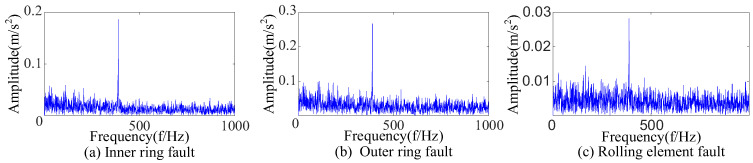
The envelope spectrum after adding noise.

**Figure 8 sensors-23-09441-f008:**
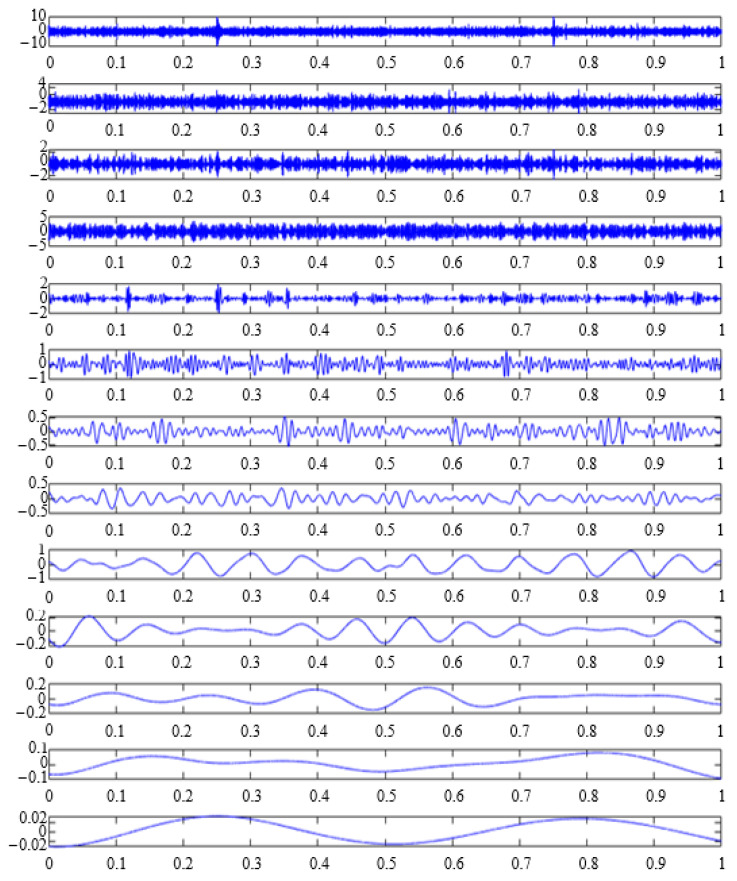
IMFs based on CEEMDAN.

**Figure 9 sensors-23-09441-f009:**
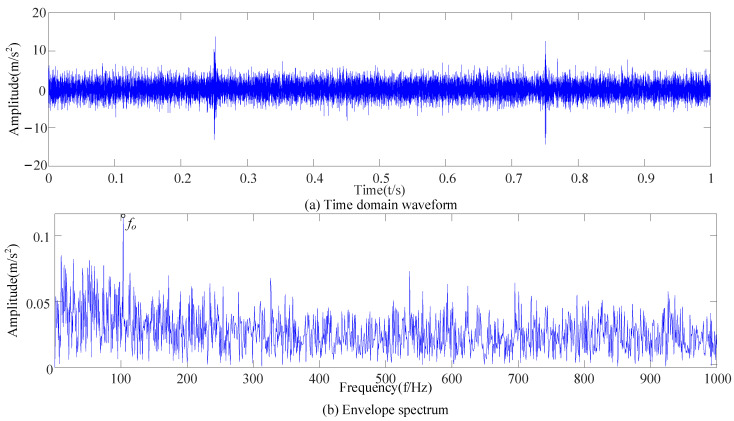
Reconstructed signal time domain and envelope spectra.

**Figure 10 sensors-23-09441-f010:**
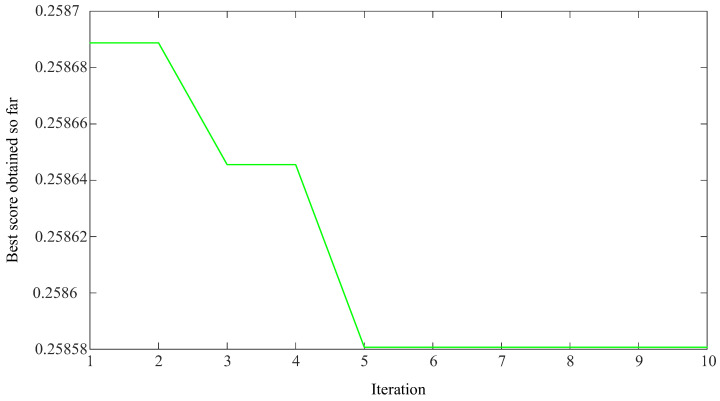
Iteration graph of VMD parameter optimization based on SSA.

**Figure 11 sensors-23-09441-f011:**
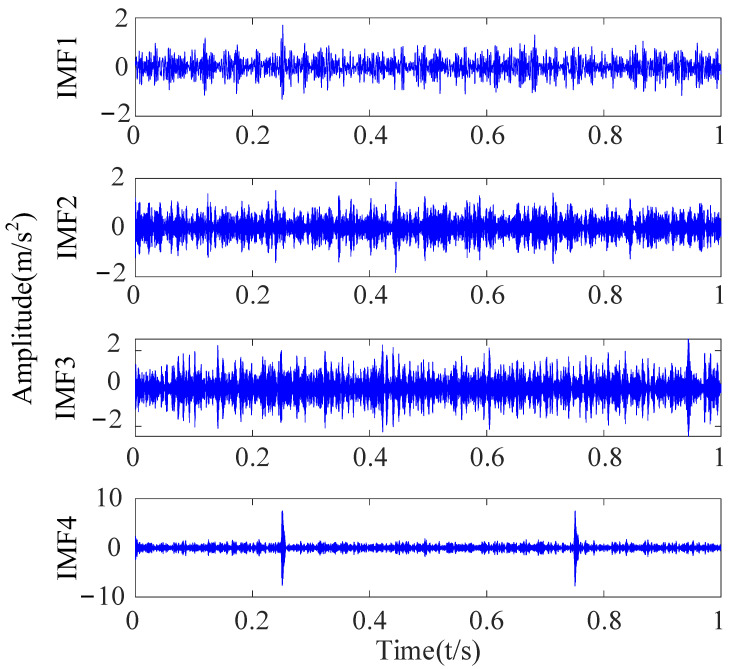
Time-domain diagrams of decomposed signal based on VMD.

**Figure 12 sensors-23-09441-f012:**
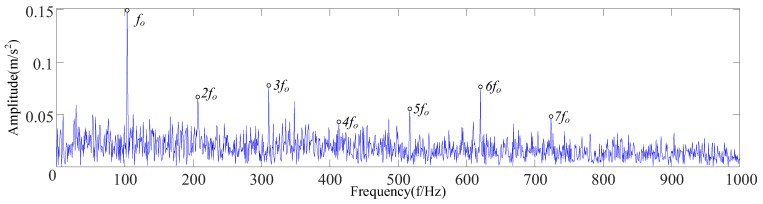
Analysis diagram of envelope demodulation for IMF3.

**Figure 13 sensors-23-09441-f013:**
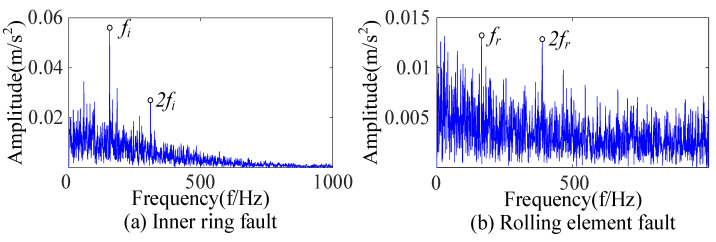
Envelope spectra of bearings with an inner-ring fault and a rolling-element fault.

**Figure 14 sensors-23-09441-f014:**
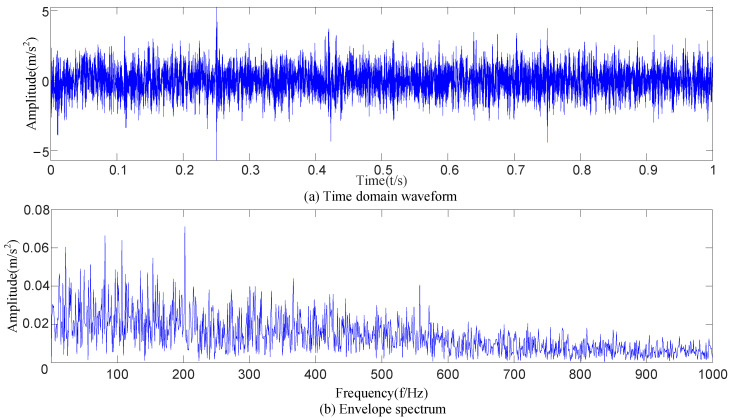
Time-domain diagram and envelope spectrum of reconstructed signal.

**Figure 15 sensors-23-09441-f015:**
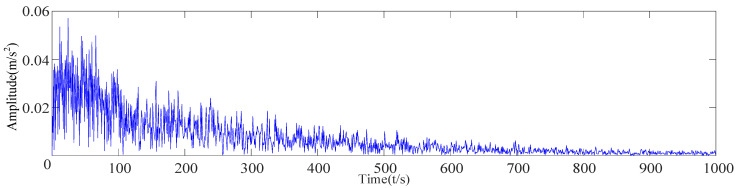
Envelope spectrum based on envelope entropy.

**Figure 16 sensors-23-09441-f016:**
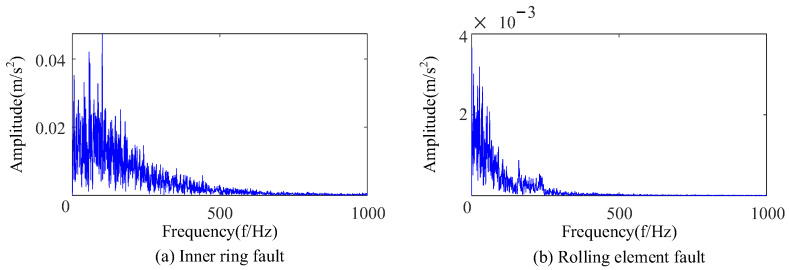
Envelope spectra of inner-ring fault and rolling-element fault based on envelope entropy.

**Figure 17 sensors-23-09441-f017:**
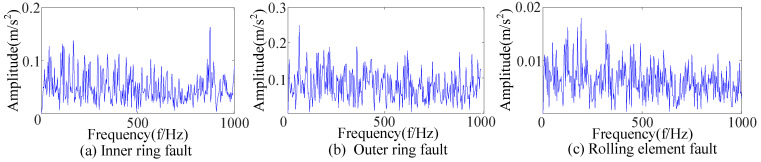
Envelope spectra of three types of bearing faults based on EEMD-WTD.

**Figure 18 sensors-23-09441-f018:**
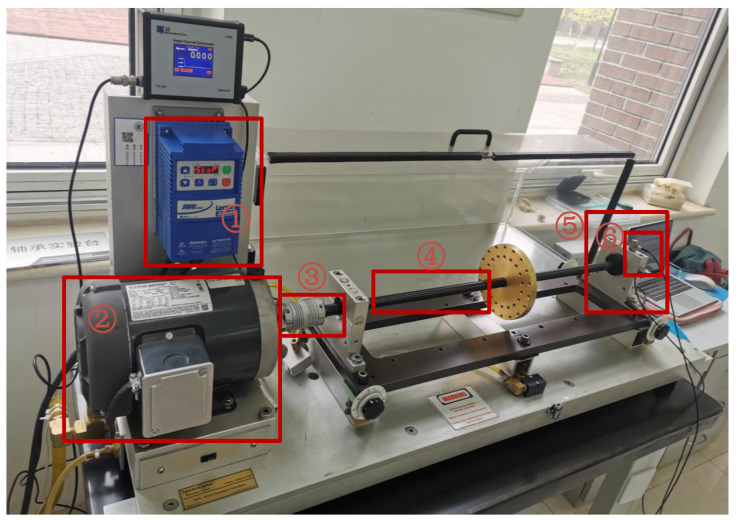
Rolling bearing test bed. ① Motor driver, ② three-phase AC motor, ③ coupling, ④ rotating shaft, ⑤ faulty bearing and ⑥ its foundation.

**Figure 19 sensors-23-09441-f019:**
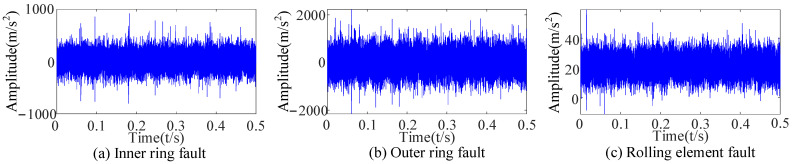
Time-domain diagrams of signals with added noise.

**Figure 20 sensors-23-09441-f020:**
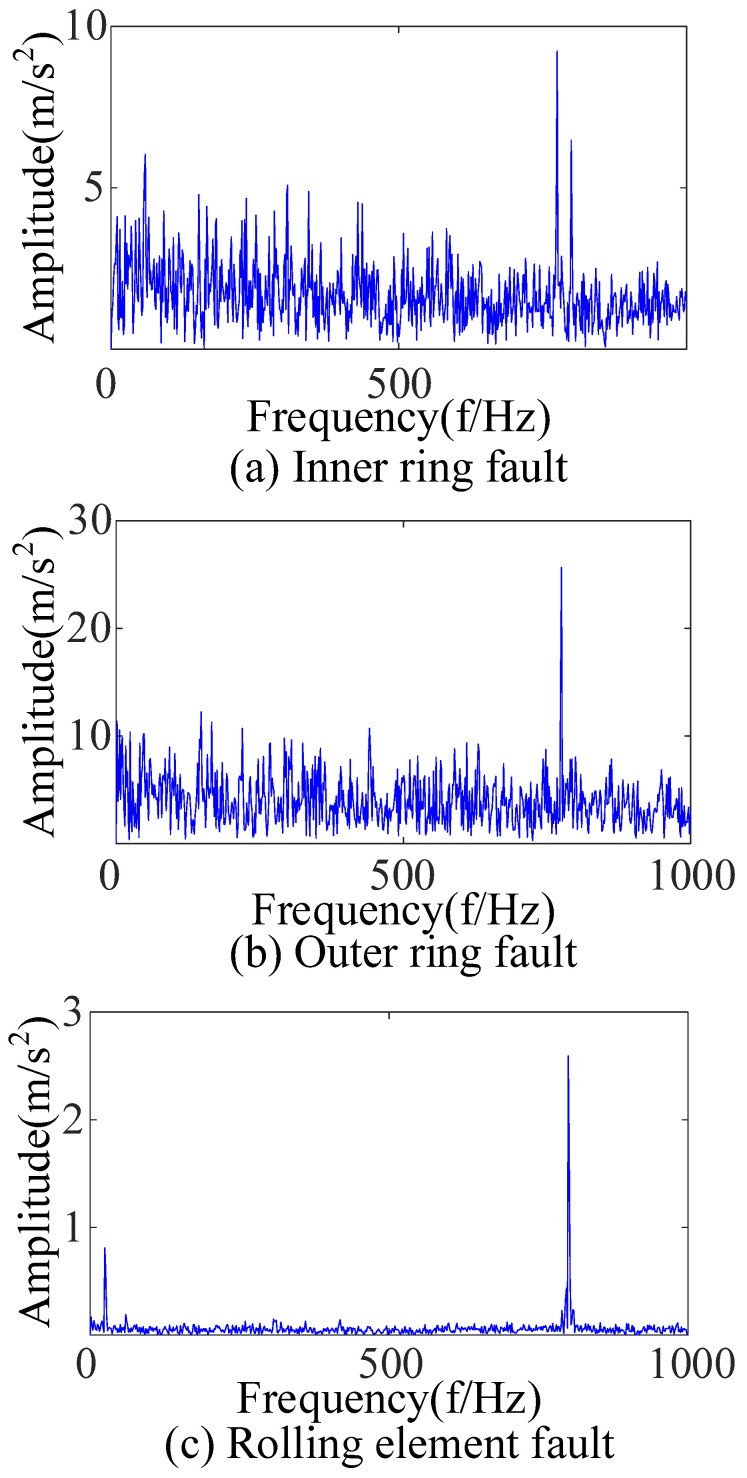
Envelope spectra of signals with added noise.

**Figure 21 sensors-23-09441-f021:**
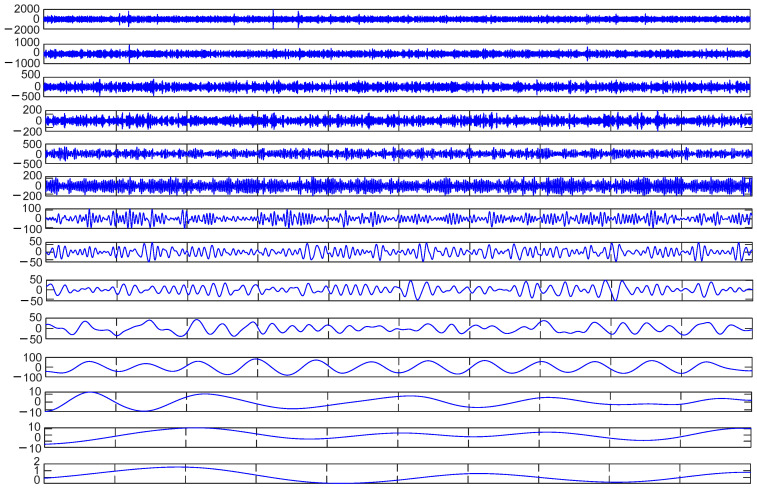
Signal decomposition diagram based on CEEMDAN.

**Figure 22 sensors-23-09441-f022:**
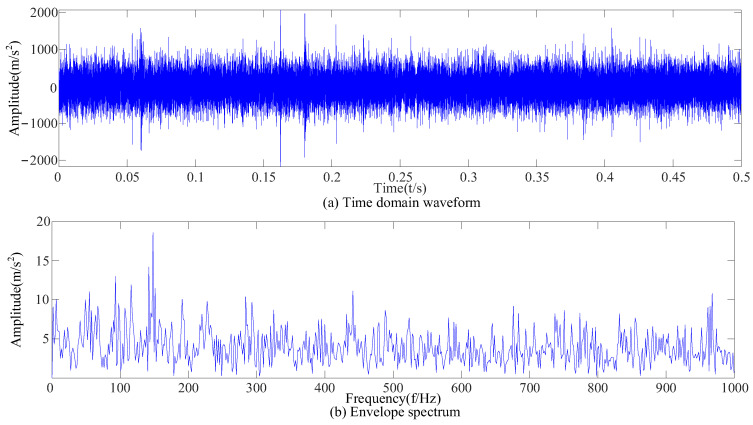
Reconstructed signal diagram.

**Figure 23 sensors-23-09441-f023:**
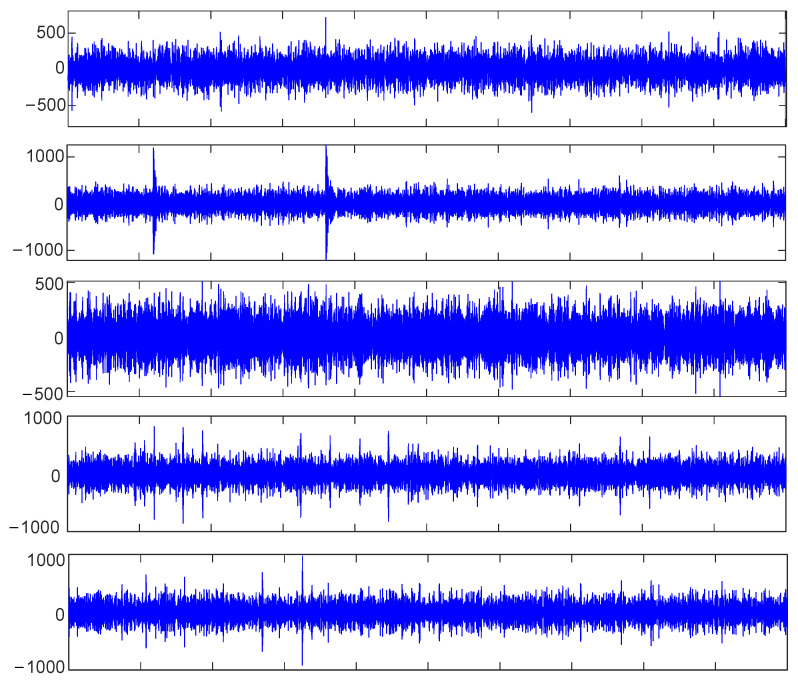
Exploded component diagram based on VMD.

**Figure 24 sensors-23-09441-f024:**
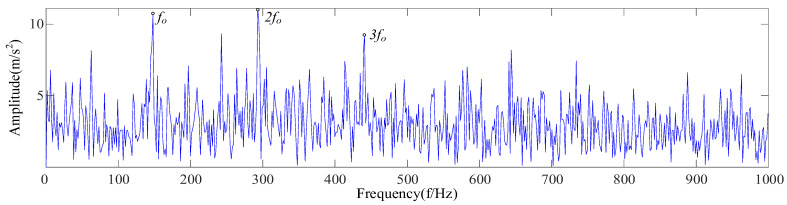
Analysis diagram of envelope demodulation for IMF5.

**Figure 25 sensors-23-09441-f025:**
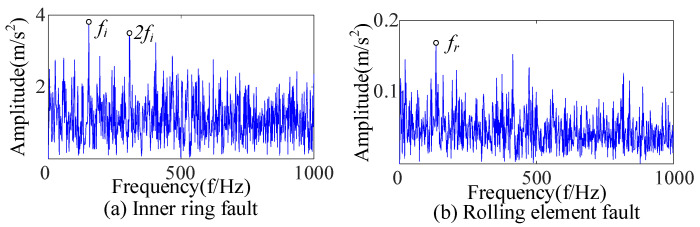
Envelope spectra of bearings with inner-ring fault and rolling-element fault.

**Figure 26 sensors-23-09441-f026:**
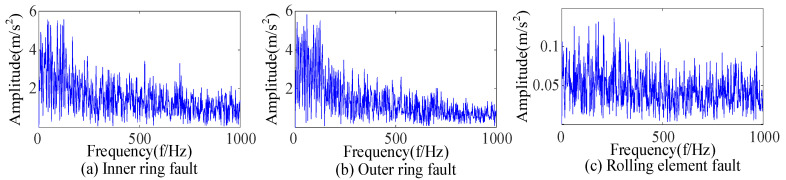
Envelope spectra based on envelope entropy.

**Figure 27 sensors-23-09441-f027:**
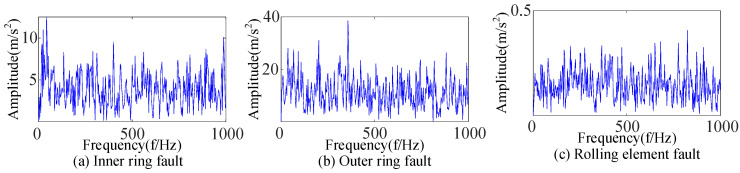
Envelope spectra based on EEMD-WTD.

**Table 1 sensors-23-09441-t001:** Parameters of interference noise.

*R* _1_	*r* _1_	*λ*	*f* * _o_ *	*A* _1_	*f_r_*	*θ* _1_	*B* _1_	*R*	*θ* _2_
1.5	0.12	400	4500	0.2	12.5	π/2	0.64	16	π/2

**Table 2 sensors-23-09441-t002:** Envelope kurtosis of signals for each IMF component.

	X	IMF1	IMF2	IMF3	IMF4	MF5	IMF6	IMF7	IMF8	IMF9	IMF10	IMF11	IMF12	IMF13
K_F_	3.29	16.05	3.53	3.81	2.98	8.35	3.14	3.14	2.73	2.52	2.31	1.99	2.10	3.66

**Table 3 sensors-23-09441-t003:** Envelope spectrum kurtosis of each component.

	IMF1	IMF2	IMF3	IMF4
K_H_	4.46	3.44	19.42	5.04

**Table 4 sensors-23-09441-t004:** CCC under different faults based on envelope entropy.

	Inner 1	Inner 2	Inner 3	Out 1	Out 2	Out 3	Roll 1	Roll 2	Roll 3
P_c_	0.0304	0.0296	0.0325	0.0041	0.0256	0.0129	0.0021	0.0025	0.0147

**Table 5 sensors-23-09441-t005:** CCCs for different faults based on EEMD-WTD.

	Inner 1	Inner 2	Inner 3	Out 1	Out 2	Out 3	Roll 1	Roll 2	Roll 3
P_c_	0.0350	0.0316	0.0332	0.0307	0.0363	0.0350	0.0181	0.0207	0.0193

**Table 6 sensors-23-09441-t006:** CCCs for different faults based on the method proposed in this paper.

	Inner 1	Inner 2	Inner 3	Out 1	Out 2	Out 3	Roll 1	Roll 2	Roll 3
P_c_	0.5128	0.5264	0.5083	0.4649	0.4502	0.4613	0.7201	0.6843	0.7014

**Table 7 sensors-23-09441-t007:** Envelope kurtosis of signal with added noise and each IMF component.

	X’	IMF1	IMF2	IMF3	IMF4	MF5	IMF6	IMF7	IMF8	IMF9	IMF10	IMF11	IMF12	IMF13	IMF14
KH	3.13	9.49	4.31	3.93	3.39	2.76	2.7	3.77	3	3.54	2.57	5.31	2.51	1.89	2.88

**Table 8 sensors-23-09441-t008:** Component envelope spectrum kurtosis.

	IMF1	IMF2	IMF3	IMF4	IMF5
K_F_	2.67	2.74	3.45	2.88	5.01

**Table 9 sensors-23-09441-t009:** CCCs for different faults based on the three methods.

P_c_	Inner 1	Inner 2	Inner 3	Out 1	Out 2	Out 3	Roll 1	Roll 2	Roll 3
CEEMDAN-VMD	0.1993	0.1809	0.2103	0.2425	0.2637	0.2492	0.3821	0.3712	0.3697
Envelope entropy	0.0833	0.0895	0.0927	0.0819	0.0763	0.0771	0.0083	0.0132	0.0096
EEMD-WTD	0.0125	0.0296	0.0249	0.0317	0.0322	0.0284	0.0181	0.0329	0.0237

## Data Availability

The data presented in this study are available on request from the corresponding author. The data are not publicly available due to [privacy].

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
