# Peer review of "Fault Feature Extraction Method for Rolling Bearings Based on Complete Ensemble Empirical Mode Decomposition with Adaptive Noise and Variational Mode Decomposition"

_sensors, 2023, doi:10.3390/s23239441_

Round 1
Reviewer 1 Report
Comments and Suggestions for Authors
In this paper, a fault features extraction method based on CEEMDAN and SSA-VMD is proposed to solve the difficulty of fault feature extraction for rolling bearings. Overall, the paper is well written and organized with a proper length. The contributions as well as the quality are both good. In addition, there are some points that are not very clear and should be addressed in the revised version:
1. Full name of abbreviation CEEMDAN, VMD, SSA, and et al should be added in Abstract section.
2. The innovation of this paper is not clear and it is difficult for readers to understand the main contributions of this paper. This part should be added in Introduction section.
3. The description of the existing work should be shorter in Introduction section. Furthermore, more descriptions of the proposed method are needed.
4. Please add the source of the public data sets of Case Western Reserve University (CWRU) in Experiment section, a public link or reference?
5. How to guarantee the noise is filtered out, while the weak fault features are retained at the same time. Please give some explanations.
6. Multivariable statistics is an important research issue which is widely applied on incipient fault diagnosis and weak fault features extraction of rotating machinery. The authors should supplement some results on this aspect, for example the following references had given significant design results:
[1] Deep PCA-Based Incipient Fault Diagnosis and Diagnosability Analysis of High-Speed
Railway Traction System via FNR Enhancement. Machines, 2023, 11(4): 475. Comments on the Quality of English Language
Minor editing of English language required
Author Response
Dear reviewer,
Thank you for your valuable suggestions and decision to revise our manuscript. We thank
you for your very constructive comments. According to the review, the manuscript has been
carefully and fully revised, with detailed explanations and corrections listed point by point
below. All changes made in the manuscript are annotated.
If there are any other problems with our manuscript, please feel free to let us know.
We look forward to hearing from you.
Yours sincerely,
Hongjiang Li

Reviewer 2 Report
Comments and Suggestions for Authors
This paper proposes a fault feature extraction method for rolling bearing based on CEEMDAN and VMD.
The originality of this paper is reducing reconfiguration signal errors, CEEMDAN added positive and negative adaptive Gaussian white noise in each stage of decomposition, averaged the decomposed IMF component and selected components by envelope kurtosis.
This method is not clear from the experimental study.
The contribution of authors is not well explained.
Therefore, it is easy to see that the article needs changes for publication.
Some remarks:
1) Figure 2 and Figure 18 are ineligible and need to be improved.
2) Lines 258 and line 262 are missing equations 17 and 18.
3) Include a one-line space between table 1 and Figure 5; table 2 and figure 9; table 3 and figure 12; table 5 and figure 24.
4) References are not standardized. For example, line 474 contains the year (2018), different from line 481.
5) The conclusion must be written in paragraphs and not in numbered steps.
Author Response

(The authors gave the same response as above.)

Reviewer 3 Report
Comments and Suggestions for Authors
All of the remarks (major and specific) are attached in the pdf file.

Author Response

(The authors gave the same response as above.)

Round 2
Reviewer 1 Report
Comments and Suggestions for Authors
In a general way most of my comments were answered by the authors. My overall opinion about this paper is quite good. The manuscript is well written and acceptable for publishing. However, the authors' Response 6 may have no relationship with the original Comment 6 as following:
6. Multivariable statistics is an important research issue which is widely applied on incipient fault diagnosis and weak fault features extraction of rotating machinery. The authors should supplement some results on this aspect, for example the following references had given significant design results:
[1] Deep PCA-Based Incipient Fault Diagnosis and Diagnosability Analysis of High-Speed Railway Traction System via FNR Enhancement. Machines, 2023, 11(4): 475.
Please have a check.
Comments on the Quality of English LanguageMinor editing of English language required
Author Response
Dear reviewer,
Thank you for your valuable suggestions and decision to revise our manuscript.
I'm sorry that I didn't understand your opinion correctly in the last revision. This time we have added a discussion of multivariable statistics to the introduction. And added annotations to the manuscript. If there are any other problems with our manuscript, please feel free to let us know.
We look forward to hearing from you.
Yours sincerely,
Hongjiang Li
Reviewer 3 Report
Comments and Suggestions for Authors
Thank you very much for providing the changes. All of my remarks on the original manuscript have been taken into account and incorporated into the content.
I have no other remarks on the current form of the manuscript.
Author Response
Dear reviewer,
Thank you very much for all your valuable comments on this paper. If there are any other problems with our manuscript, please feel free to let us know.
Yours sincerely,
Hongjiang Li